# Age, period, and cohort effects of *Clonorchis sinensis* infection prevalence in the Republic of Korea: Insights and projections

**Sung-mok Jung**[1]*, **Heewon Kang**[2], **Bong-Kwang Jung**[3], **Sejin Ju**[4], **Jung-Won Ju**[5], **Myoung-Ro Lee**[5], **Jong-hun Kim**[6], **Sung Hye Kim**[4]*

**1** Carolina Population Center, University of North Carolina at Chapel Hill, Chapel Hill, North Carolina, United States of America, **2** Institute of Health and Environment, Seoul National University, Seoul, Republic of Korea, **3** MediCheck Research Institute, Korea Association of Health Promotion, Seoul, Republic of Korea, **4** Center of Global Health Practice, Institute of Health and Society, Hanyang University College of Medicine, Seoul, Republic of Korea, **5** Division of Vectors and Parasitic Diseases, Korea Diseases Control and Prevention Agency, Osong, Republic of Korea, **6** Department of Social and Preventive Medicine, Sungkyunkwan University School of Medicine, Suwon, Republic of Korea

\* sungmok@ad.unc.edu (S-mJ); sunghyekim@hanyang.ac.kr (SHK)

**Data Availability Statement:** Reports on the age-specific prevalence of C. sinensis infection for the years 1981 (https://dl.nanet.go.kr/search/searchInnerDetail.do? S&controlNo=

## Abstract

### Background

With decades of containment efforts, the prevalence of *C. sinensis* infection in Korea has shown a declining trend. However, well-tailored intervention assessments remain challenging, particularly when considering the potential impacts of cohort variations in raw freshwater fish consumption behavior, a major transmission route to humans, on this observed decline.

### Methodology

We applied an age-period-cohort modeling approach to nationally representative *C. sinensis* infection prevalence data from 1981–2012 in Korea to assess age, period, and cohort effects on its secular trend and to project the age-stratified prevalence up to 2023.

### Principal findings

Our analysis suggests that both cohort and period effects have substantially contributed to the declining prevalence of *C. sinensis* infection in Korea. Age-stratified projections up to 2023 suggest a decline in prevalence across all age groups, while those aged over 40 are anticipated to maintain prevalences above the elimination threshold of 1%.

### Conclusions

Our study highlights the importance of incorporating cohort effects into intervention assessments aimed at controlling *C. sinensis* infection. The effectiveness of interventions remains evident in Korea despite adjusting for the cohort effect. This approach, applicable to other endemic countries, would provide valuable insights for intervention assessments and inform future public health planning to eliminate *C. sinensis* infection.

MONO1199319953), 1986 (https://dl.nanet.go.kr/search/searchInnerDetail.do? S&controlNo=MONO1199202313), 1992 (https://dl.nanet.go.kr/search/searchInnerDetail.do? S&controlNo=MONO1199318513), 1997 (https://dl.nanet.go.kr/search/searchInnerDetail.do? S&controlNo=MONO1199800554), 2004 (https://library.nih.go.kr/ncmiklib/mlib/mlibViewVolume.do?searchSubTarget=volume&bibctrlno=1398486), 2012 (https://www.mohw.go.kr/board.es?mid=a10411010100&bid=0019&tag=&act=view&list_no=337123), and the age-specific prevalence data in high-endemic regions from 2013–2021 (doi:10.3347/kjp.2022.60.3.187) are publicly available. Individual-level data for the 2012 survey and the prevalence data in high-endemic regions for 2021 are not publicly available, in accordance with the policy of the KDCA protected by Korean law. Requests for access to this data can be directed to the Division of Vectors & Parasitic Diseases, Bureau of Infectious Disease Diagnosis Control at the Korea Disease Control and Prevention Agency (through the website: https://www.open.go.kr). Each request will be reviewed by the KDCA. The replication code is available on the GitHub repository https://github.com/SungmokJung/Clonorchis_sinensis_APC_Korea.

**Funding:** This work was supported by the Centers for Disease Control and Prevention Safety and Healthcare Epidemiology Prevention Research Development programme (200-2016-91781 to S-mJ), and by Hanyang University College of Medicine (HY-2020000000000495 to SHK). This study was also supported by the Korea Disease Control and Prevention Agency (KDCA 6331-311, 2022). The funders had no role in study design, data analysis, decision to publish, or manuscript preparation.

**Competing interests:** I have read the journal's policy and the authors of this manuscript have the following competing interests: J-hJ and M-RL are affiliated with the Division of Vectors and Parasitic Diseases, Korea Disease Control and Prevention Agency. The findings and conclusions in this paper are those of the authors and do not necessarily represent the official position of the Korea Disease Control and Prevention Agency. Any use of trade, firm, or product names is for descriptive purposes only and does not imply endorsement by the Korean government.

## Author summary

*Clonorchis sinensis* (*C. sinensis*) infection, a neglected chronic parasitic infection, continues to be a significant public health concern with severe implications. Despite ongoing containment efforts leading to a decline in prevalence in Korea, assessing interventions remains challenging, due to variations in infection risk across different birth cohorts, driven by generational differences in raw freshwater fish consumption. We applied the age-period-cohort modeling approach to nationally representative *C. sinensis* infection prevalence data from 1981–2012 in Korea. Our analysis showed both cohort and period effects have significantly contributed to the observed decline in the prevalence of *C. sinensis* infection in Korea throughout the study period. Notably, the effectiveness of interventions remains evident even after adjusting for the cohort effect. Projections of age-stratified prevalence up to 2023 suggest a downward trend across all age groups. However, those aged over 40 are expected to maintain a prevalence exceeding 1%, highlighting the imperative for ongoing sustained efforts toward disease elimination. The age-period-cohort modeling framework offers broad applicability in other liver fluke endemic countries, particularly where data on host reservoirs beyond humans is limited. This approach provides valuable insights for future public health planning and resource allocation, aiming to mitigate the disease burden in affected regions.

## Introduction

*Clonorchis sinensis* (*C. sinensis*), a prevalent intestinal helminth primarily endemic to the East Asian region, including the Republic of Korea (hereafter "Korea"), China, and northern Vietnam [1–3], poses significant public health threats in this region. This food-borne trematode completes its life cycle through three reservoir hosts—freshwater snails, freshwater fish, and humans (or other carnivorous mammals)—with the consumption of raw or undercooked freshwater fish infected with *C. sinensis* metacercariae serving as the primary mode of transmission to carnivorous mammals, including humans [4]. The repercussions of *C. sinensis* infection can be severe, including damage to the liver and biliary systems, which may lead to fatal cholangiocarcinoma. While treatment of the disease has progressed with the introduction of praziquantel in the early 1980s [5], the global burden of *C. sinensis* remains substantial, with an estimated 15 million individuals infected worldwide and over 200 million people at risk of contracting the parasite as of 2004 [6].

Such public health threats posed by *C. sinensis* are particularly concerning in Korea, where its positivity rates have surpassed those of all other intestinal parasitic infections in the country over the past decade [7]. To control the spread of the parasite, the Korean government initiated the first nationwide intestinal parasite elimination program in the late 1960s. This initiative, lasting until 1994, involved health education campaigns and regular fecal screenings (twice per year) primarily targeted at the school-age population, along with the widespread allocation of praziquantel since 1983 [8,9]. With these proactive measures, involving around 10 million individuals annually, the prevalence of most parasitic infections in Korea significantly decreased to below 1%. However, the effectiveness of these measures in *C. sinensis* infection was limited, with a national prevalence of 2.2% reported in 1992, the highest among all parasitic infections [10]. It was especially notable in regions along the five main river streams, where the consumption of raw freshwater fish was prevalent, resulting in prevalences exceeding 25% in 2005 [2]. In response, the government reinstated the elimination program in 2005,

with a renewed focus on reducing the infection rate in these endemic regions through active screening campaigns and follow-up antiparasitic treatment [8]. Despite these sustained efforts and a further decline in national prevalence to 1.9% by 2012, the country continues to face areas with high endemicity, where *C. sinensis* infection comprises more than 70% of the total positivity rate for all reported intestinal parasitic infections [7].

Alongside these elimination programs, the Korean government has conducted periodic nationwide surveys every 5–8 years since 1971 to monitor the major intestinal parasitic infections and guide control measures. Despite these initiatives, attaining a comprehensive understanding of the epidemiology of *C. sinensis* infection remains a formidable challenge in Korea. The intervals of 5–8 years between the surveys have hindered the timely acquisition of epidemiologic profiles of the disease during this period and precise assessments of intervention impacts. In such instances, statistical modeling approaches that incorporate retrospective time-series trends or environmental factors driving the spread of pathogens are often utilized to bridge these gaps and evaluate the effectiveness of intervention strategies [11,12]. However, it is crucial to note that such methods may introduce biases in intervention assessments, particularly in the context of *C. sinensis* infection, where the risk of infection may vary by birth cohort (referred to as "cohort effect" [13]). Previous studies, albeit limited to cross-sectional analysis, suggested differing raw freshwater fish consumption behavior among birth cohorts in Korea, with higher rates observed in older generations and substantially lower rates in younger generations (S1 Fig) [7,14]. Due to these cohort variations, the infection risk of *C. sinensis* is lower in younger generations, thereby, their natural replacement of higher-risk generations through aging might contribute to a decrease in infections over time, irrespective of intervention effects. Neglecting this cohort effect in conventional approaches could overestimate intervention effects, potentially leading to misguided future public health policies and resource misallocation.

Age-period-cohort (APC) model stands as a useful tool for elucidating the longitudinal patterns of epidemiological events and disentangling the dynamic effects of age, period, and cohort on these trends [15–17]. In this framework, age effects reflect the physiological changes individuals undergo across their lifespan, while period effects signify simultaneous shifts affecting all age groups, often precipitated by specific events such as interventions. The cohort effects capture the collective influence of age and period effects, revealing variations among individuals born in the same year. Given the potential consolidation of these effects in shaping the observed trend of *C. sinensis* infection prevalence in Korea, it is crucial to quantitatively assess the individual contribution of each effect, which enables well-tailored intervention assessments while adjusting for potential confounding.

To better understand the observed trend of *C. sinensis* infection prevalence and retrospectively evaluate implemented interventions in Korea, we applied the APC modeling approach to nationally representative time-series data from 1981–2012. We then extend this framework by projecting the age-stratified prevalence up to 2023 to illuminate the unobserved gap arising from the survey interval and inform future public health strategies.

## Methods

### Ethics statement

This study was approved by the Institutional Review Board of Korea Association of Health Promotion (2022-HR-009) and Sungkyunkwan University School of Medicine (No. 2024-03-036), and the participant consent was waived by the ethics committees as the data involved routinely collected surveillance data that was processed anonymously at all stages.

## Epidemiological data

Age-stratified prevalence of *C. sinensis* infection in Korea from 1981–2012 (specifically 1981, 1986, 1992, 1997, 2004, and 2012) was extracted from national surveys on the prevalence of intestinal parasitic infections conducted by the Korea Disease Control and Prevention Agency (KDCA), with the last survey conducted in 2012 [7,18–22]. To ensure a representative sample, the total number of survey regions was strategically allocated into 16 provinces in Korea, factoring in urban/rural distinctions and positivity rates of three prevalent intestinal parasites— *C. sinensis*, heterophyids, and pinworm [10]. Allocated numbers of survey regions were then sampled from enumeration districts utilized in the National Population and Housing Census [23], proportionate to the number of households. Within each sampled region, participant households were randomly selected. Further details on the sampling method, including minor changes across survey iterations, are described in S1 Table. The prevalence data from 1981– 2004 was collected from published survey reports, while that for 2012, along with details on freshwater fish consumption behavior (exclusively available in 2012), was directly extracted from the individual-based survey data. All data were categorized into five age groups with 10-year bands: 20–29, 30–39, 40–49, 50–59, and 60 years and over (60+). Individuals under 19 were excluded due to their negligible prevalence of C. sinensis throughout the study period, and those aged 60+ were combined into a single group given their relatively small sample size. Age-stratified estimated population size was retrieved from the Korean Statistical Information Service (KOSIS) website [24].

## Age-period-cohort modeling

To examine age, period, and cohort effects on the secular trend of *C. sinensis* infection in Korea, we employed the APC modeling approach on the time-series prevalence data from 1981–2012. We assumed that the observed prevalence in each survey iteration acts as a proxy for the incidence rates of *C. sinensis* infection in Korea, given the widespread implementation of active screening campaigns and follow-up antiparasitic treatment in Korea throughout the survey years [8]; thus, following a Poisson distribution. We then modeled prevalence $\lambda$ at age group $a$ in survey calendar year $p$ for those in birth cohort $c$ as follows:

$$log(\lambda(a,p)) = f(a) + g(p) + h(c), \tag{1}$$

where cohort $c$ is defined as the difference between the start of the age group intervals and the calendar year of surveys ($c = p - a$). To address a non-identifiability issue arising from the linear dependency between these effects, the age-period (or age-cohort) model was first fitted, and then the logarithm of the fitted values was employed as an offset variable in a cohort effect (or period effect) model [25]. The APC modeling was performed using *R* package '*Epi*' [26], referencing 1981 as the period and 1916 as the cohort. Models with different combinations of age, period, and cohort, along with numbers of B-spline knots for each of the terms were selected based on the Akaike Information Criterion (AIC).

## Projecting the prevalence of *C. sinensis* infection

To project the age-stratified prevalences of *C. sinensis* infection in Korea, we employed Bayesian age-period-cohort analysis with the Integrated Nested Laplace Approximations, as implemented in the *R* package '*BAPC*' [27]. To consider changes in population structure over the survey period, observed prevalences among tested samples were converted into population-level estimates based on the age-stratified population size, while uniform population sizes were applied over every 5-year interval due to the data availability. Annual age-stratified prevalences

were projected up to 2023, covering the next 10 years beyond the last available data (2012), based on the national prospective population projections [24] and the posterior distribution of each effect obtained by fitting the model to the retrospective time-series data.

To assess the plausibility of our projections, we compared them with the observed *C. sinensis* infection prevalence from 2013–2021 (covering the period from the start projections to the most recent data available) among residents in high-endemic areas of intestinal parasitic infections across Korea. This time-series data, sourced from a published paper (for 2013–2020) [28] and supplemented with non-public data from the KDCA (for 2021) [29], comprises positive rates of the disease obtained during annual screening campaigns conducted in 30–40 cities, primarily located near the five main river streams in Korea (S2 Table). While this data covers a more recent period than our main data (1981–2012), it was used exclusively for validating the overall trend of projections due to (i) its limited timeframe and geographical area, which restrict accurate assessments of the cohort effects [30] and (ii) the inability to combine it with our main data, given the differing sampling design and participant group characteristics (i.e., nationally representative versus high-endemic region residents). All analysis was performed in *R* version 4.3.2.

## Results

### Trends in the time-series prevalence of *C. sinensis* infection in Korea

Up to the latest national survey conducted in 2012, the temporal pattern of *C. sinensis* infection prevalence in Korea has shown a slightly decreasing trend, yet a notable surge was observed in 2004, accounting for 80% of the positivity rate against all parasitic infections (Fig 1A). During the 1980s, high prevalences exceeding 4% were prominent across those aged 30–59 (Fig 1B). However, a distinct reduction has been evident since the 1990s, particularly in the 30s, where the prevalence dropped to around 2% (Fig 1C). Despite this decline, prevalences among those

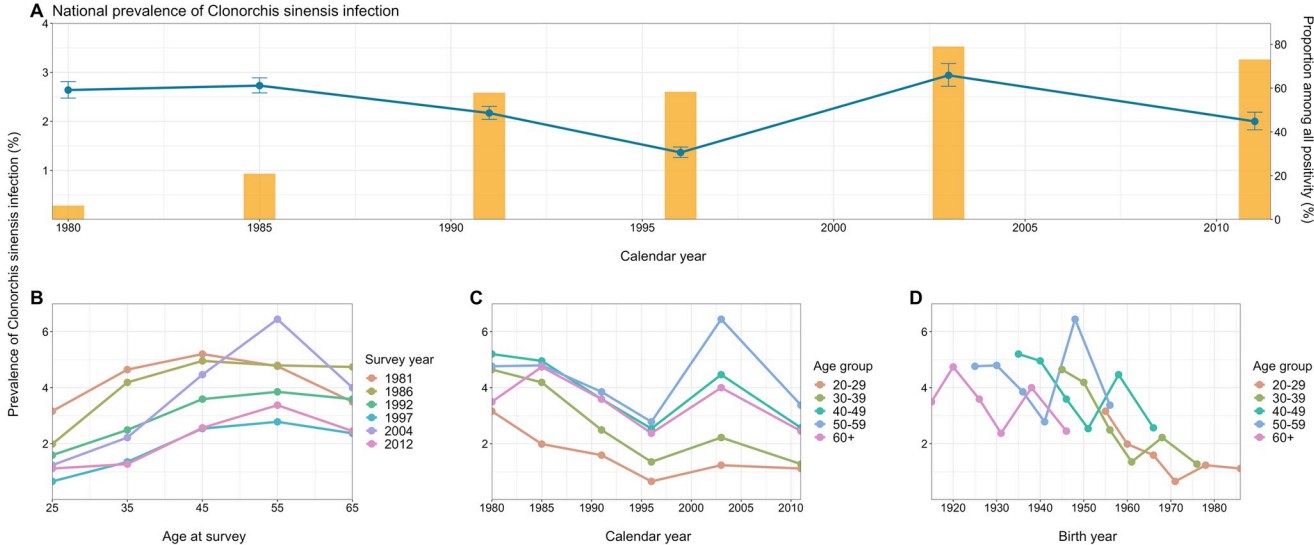

**Fig 1. Overview of *Clonorchis sinensis* infection prevalence in the Republic of Korea, 1981–2012.** (**A**) National-level prevalence of *Clonorchis sinensis* infection per 100 in the Republic of Korea from 1981–2012, extracted from national surveys on intestinal parasitic infections. Blue dots and error bars represent the time-series prevalence of *Clonorchis sinensis* infection, and yellow bars indicate the proportion of *Clonorchis sinensis* infection among positivity against all parasitic infections in each survey. (**B–D**) Each panel presents the observed prevalence (**B**) by age at survey (the median age of each corresponding age group, with 65 designated for those aged over 65) and survey year, (**C**) by calendar year and age group, and (**D**) by birth year and age group, respectively.

aged over 50 have consistently remained higher than in other age groups, albeit lower than the peak seen in the 1980s. This diminishing trend in younger age groups is also depicted in the observed prevalence by birth cohort and age group (Fig 1D), suggesting a potential cohort effect on the observed secular trend in *C. sinensis* infection prevalence in Korea.

The age- and sex-stratified prevalence observed in the 2012 survey showed a gradual increase, with higher values in older age groups and among men within each age group (S1A Fig). This trend mirrored the raw freshwater fish consumption behavior pattern by age group and sex reported in the same survey, which is acknowledged as the primary transmission route to humans (S1B Fig). This association was further evident in region-stratified values (S2 Fig), showing a significantly positive correlation (Pearson's correlation coefficient: 0.23, 95% confidence interval (CI): 0.10, 0.36).

## Age-period-cohort analysis

Using the model with age, period, and cohort effects, selected as the best-fitted model (S3 Table), the prevalence of *C. sinensis* infection showed a gradual increase with age, reaching its highest estimates in those aged 50–59 (Fig 2A). The estimated prevalence ratio by cohort effect, adjusted for estimated age and period effects, indicated a substantial decrease in the risk of *C. sinensis* infection as the birth cohort year increased (Fig 2B). Specifically, in comparison to the reference group (individuals born in 1916), the estimated risk was 0.21 (95% CI: 0.14, 0.31) times lower in those born in 1979. While a slightly higher risk, but insignificant, was estimated among those born in 1987, it could be mainly attributed to the limited samples available from the corresponding birth cohort across all surveys. The estimated prevalence ratio driven by period effect generally declined over the survey years (Fig 2C). This decreasing trend, after adjusting for the contribution of the cohort effect, suggests the effectiveness of interventions in Korea, including sustained non-pharmaceutical measures and the widespread application of praziquantel since 1983 [31]. Notably, fluctuations observed in 1997 and 2004, coincided with the suspension and subsequent reinstation of the nationwide eradication campaign in Korea in 1995 and 2005, respectively [32,33].

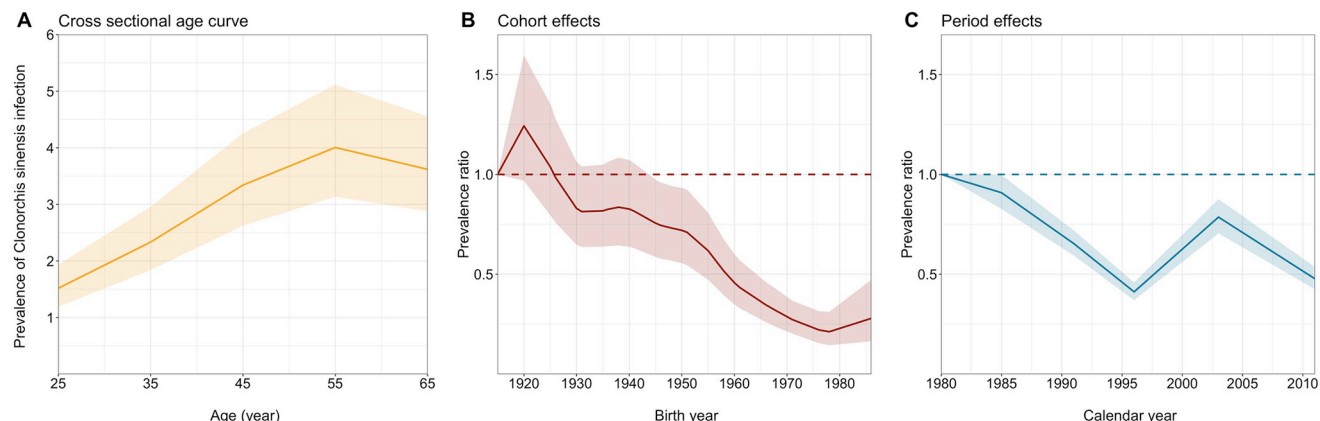

**Fig 2. Estimated age, cohort, and period effects in the observed *Clonorchis sinensis* infection prevalence in the Republic of Korea, 1981–2012. (A)** Estimated prevalence of *Clonorchis sinensis* infection by age, with each age group represented by its median age. (**B**) Estimated prevalence ratio by birth year, and (**C**) by calendar year, with 1916 and 1981 serving as reference years respectively. Dashed horizontal lines indicate the value of one (reference for the prevalence ratio). Lines and shades indicate the median of estimated outcomes from the age-period-cohort model and 95% confidence intervals.

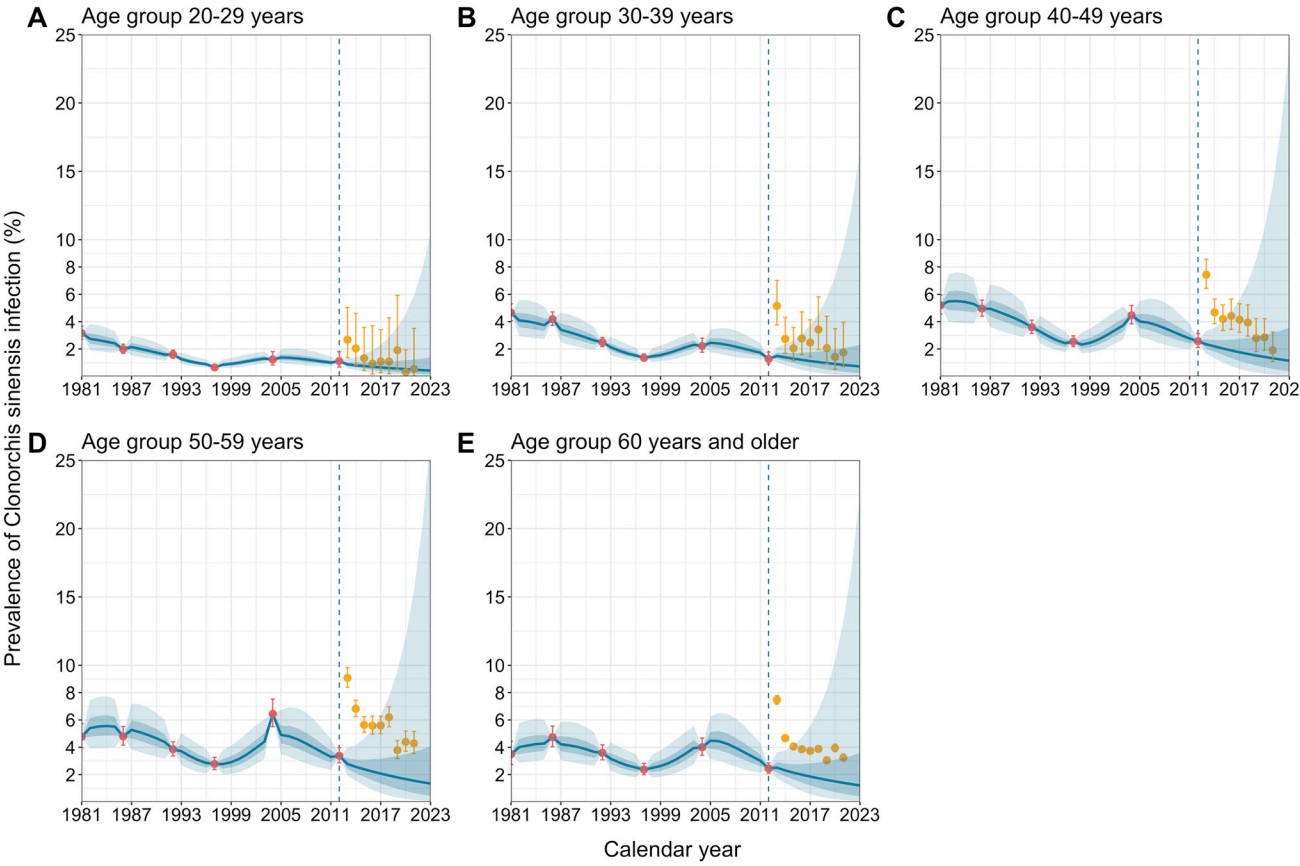

**Fig 3. Projected prevalence of *Clonorchis sinensis* infection in Korea from 2013–2023.** Projected prevalence of *Clonorchis sinensis* infection in Korea up to 2023 by age group. Blue lines indicate the median of the projected outcomes, while blue and lighter blue shades represent their 50% and 90% projection intervals, respectively. Red dots and error bars represent the observed prevalence and 95% confidence interval from national surveys on intestinal parasitic infections. Yellow dots and error bars represent the observed prevalence in high-endemic regions in Korea from 2013–2021 (serving as upper limits of the national prevalence) and their 95% confidence intervals. Dashed vertical lines show the year of the latest national survey (2012).

## Projection of future prevalence of *C. sinensis* infection

Our projection up to 2023, incorporating the age, period, and cohort effects estimated using retrospective data and prospective population projections, showed that the prevalence of *C. sinensis* infection in Korea would decrease over time across all age groups (Fig 3). Notably, the most significant decrease in the prevalence is expected in the age group 40–49, with a projected prevalence of 1.13% (90% projection interval (PI): 0.37, 3.55%) in 2023 (S4 Table), dropped from the observed prevalence of 2.57% (95% CI: 2.11, 3.12%) in the latest 2012 survey. These projected downward trends across all age groups appear to align with the overall secular trends of prevalence observed in high-endemic regions of Korea from 2013–2021, serving as upper limits for national prevalence.

## Discussions

Despite decades of efforts to control *C. sinensis* infection, its persistent transmission continues to pose a significant public health concern in Korea. Our APC analysis, utilizing nationally representative time-series data from 1981–2012, indicates both cohort and period effects have contributed to the observed declining prevalence of the disease. Furthermore, our projections

up to 2023, based on age, period, and cohort effects estimated from retrospective data, suggest a continuous decline in prevalence over time, with the median anticipated to fall below 2% across all age groups.

Our analysis quantitatively showed the role of the cohort effect in the observed decline of *C. sinensis* infection prevalence, highlighting the necessity of integrating this factor when evaluating the effectiveness of interventions. Neglecting to account for the cohort effect could potentially lead to overestimation in the assessment. However, it is important to note that this cohort effect might also be linked to the implemented interventions, particularly when it is attributed to generational variations in raw freshwater fish consumption. Since 1970, the Korean government launched comprehensive school-based health promotions (from primary to high schools) aimed at enhancing awareness about intestinal parasitic infections and their major transmission routes to humans, including the consumption of freshwater fish in raw form [8]. Consequently, the younger generation (those born in the cohort of 1952 or later) is likely to show lower engagement in consuming raw freshwater fish, whereas the older generation might sustain their dietary practices. Such generational differences in raw freshwater fish consumption might be further accentuated by the increasing adoption of Westernized diets among younger generations [34], while also being influenced by familial dietary assimilation [35]. In addition, the extent to which cohort effects contribute to the declining prevalence may vary at the regional level, depending on the ecological dynamics of reservoirs in each region (e.g., species and habitat distribution of reservoirs involved in the *C. sinensis* transmission). For instance, despite comparable levels of raw freshwater fish consumption behavior to other regions, the reported prevalence in Gangwon province was notably low (S2 Fig), attributable to the reported limited inhabitant area of the fish with *C. sinensis* infection in this region [36], as well as potentially differing ecological dynamics of the first intermediate host (snails).

In addition to the contribution of the cohort effect, our APC analysis showed a significant reduction in the risk of *C. sinensis* infection associated with the period effect. This suggests the potential effectiveness of both national-level interventions (1969–1995), including the nationwide application of praziquantel starting in 1983, and targeted interventions implemented in endemic regions (since 2005) in controlling *C. sinensis* infection in Korea [8]. This period effect could be partially attributed to improvements in the sewage system, evidenced by a notable increase in the national sewage supply coverage from 66% in 1998 to over 90% by 2007 [37]. Such enhancements are likely instrumental in limiting human-to-snail transmission of *C. sinensis*, complementing the nationwide interventions. However, the persistent downward trend in risk, following the initiation of targeted intervention in 2005, implies that active screening campaigns and follow-up pharmaceutical treatment have likely contributed to the declining prevalence of *C. sinensis* infection in Korea. Such effectiveness is further supported by the estimated increase in risk during the temporary suspension of antiparasitic campaigns (1996–2004) [32]. Similar to the cohort effect, the period effect may also vary regionally due to small-scale interventions implemented at the local public health authority level [38]. While our study primarily focused on the national level due to the limited availability of age- and region-stratified prevalence data across the study period, a more detailed assessment of interventions would be feasible with finer-grained information.

Our projections indicate that the prevalence of *C. sinensis* infection in Korea, given the estimated age, period, and cohort effects, would substantially decline across all age groups up to 2023. This downward trend is empirically supported by a well-aligned trend with observed prevalence among residents in high-endemic areas in Korea from 2013–2021, serving as upper limits of empirical prevalences at the national level (Fig 3). Nonetheless, prevalences among those over 40 years and older are projected to remain higher than 1% as of 2023, an elimination threshold conventionally targeted in other helminth infections [39]. This suggests the

ongoing necessity for the persistent implementation of interventions to achieve the elimination of the disease in Korea. However, it is worth noting that our projections might underpredict declining prevalence rates due to the inherent limitation of the statistical modeling approach (including the APC model), which may overlook the interdependent nature of infectious diseases [40]. Specifically, the reduced prevalence in younger generations can influence the overall transmission cycle of *C. sinensis* (including human-to-snail transmission); hence, it may contribute to lowering the infection risk in older generations despite their continued dietary practices of consuming raw freshwater fish.

Nevertheless, this APC modeling approach holds broad applicability to other endemic regions to *C. sinensis* infection (e.g., China, and northern Vietnam [1,3,41]), serving as an effective tool for evaluating interventions and facilitating projections (or nowcasting) between survey iterations. While mathematical modeling approaches can be applied to such projections [42], the APC modeling approach presents a notable advantage in the context of *C. sinensis* infection by not being hindered by the limited ecological and epidemiological data on reservoirs beyond humans (including fish, snails, and other carnivorous mammals), which is likely the case in most endemic countries. This scarcity complicates the accurate reconstruction of the transmission dynamics between host populations—an essential aspect of mathematical modeling approaches. Furthermore, the significance of the APC modeling approach becomes particularly apparent in endemic countries where there is significant heterogeneity in raw freshwater fish consumption behavior between generations, minimizing the risk of overestimating the effectiveness of interventions that may arise from ignoring cohort effects. In addition, while our projections do not extend into the future due to the data constraints, it should be noted that the APC modeling approach remains applicable for projecting future trajectories of the disease once updated data becomes available [15,43].

Caution should be advised when comparing our results with studies not incorporating cohort effects. Although the overall significance of intervention effects might appear consistent, their magnitude could be overestimated if the cohort effects are not explicitly considered. This is particularly relevant in the context of *C. sinensis* infection, where infection risk varies significantly by birth cohort due to generational differences in raw freshwater fish consumption. Furthermore, variations in epidemiological and ecological conditions across countries (e.g., differences in intervention target groups [44], generational freshwater fish consumption patterns, and the dynamics of intermediate hosts [45]), combined with differing statistical approaches, can further complicate direct comparisons between studies. Additionally, it is worth noting that our APC modeling approach may potentially underestimate the effectiveness of intervention specifically targeting high-risk cohort groups because adjusting for cohort effects can obscure the impact of these targeted measures. Indeed, a large-scale stool examination has been conducted in Korea since 2011 targeting residents along main river streams [8]. Although this intervention was not directed at specific cohorts, its effects might be particularly pronounced among older generations due to their higher proportion in these areas. However, in practice, disentangling the specific impacts of such targeted interventions from period and cohort effects is challenging because these interventions often coincide with broader public health initiatives (e.g., mass health campaigns aiming at changing risk behaviors of *C. sinensis* infection [46]). Moreover, even with targeted interventions, residual cohort effects due to the sustained raw freshwater fish consumption are highly likely to persist; thus, neglecting to account for the cohort effects may lead to biased estimates of intervention effectiveness and inaccurate projections of future disease trends.

Several limitations should be noted. First, our analysis relies on the assumption that the reported prevalence rate of *C. sinensis* infection is equivalent to the incidence rate, under the premise of widespread screening campaigns and subsequent follow-up treatment in Korea

throughout the study period [8]. However, the robustness of this assumption may be compromised if factors such as incomplete treatment or the accumulation of exposure to *C. sinensis* significantly affect the observed prevalence. Nonetheless, we believe that such effects are implicitly considered in our projections via the age effect (Fig 2A), which likely reflects both the accumulation of exposure and the increasing likelihood of new exposure with aging. Second, we presumed that the age, period, and cohort effects estimated from retrospective data from 1981–2012 remain constant throughout our projection period. Nonetheless, dietary practices and interventions during this period, which primarily drive period and cohort effects, are expected to undergo minimal or negligible changes. Third, our analyses did not take into account minor variations in the sampling method and fecal examination technique across survey iterations. Lastly, while our model accounts for changes in population structure, it may not fully capture more prominent shifts, particularly in high-risk regions (e.g., a gradual increase in rural-to-urban migrants among younger generations [47]), where such regional heterogeneity could impact our projections. Despite this study offering insights into the national-level situational assessment of *C. sinensis* infection, future studies applying the APC modeling approach to data from high-endemic regions with a longer observation period could provide more detailed guidance for developing future control strategies in Korea.

## Conclusions

Our study suggests the significant contributions of the cohort effect in the declining prevalence of *C. sinensis* infection in Korea, emphasizing the importance of incorporating it into intervention assessments. Even after adjusting for the cohort effect, the effectiveness of interventions remains evident. Our projections, guided by these effects, suggest that a continuous reduction in *C. sinensis* infection prevalence is anticipated across all age groups in Korea, while it may persist above the elimination threshold of 1% by 2023, particularly among older generations. This approach, applicable to other endemic countries, would provide valuable insights for intervention assessments and inform future public health strategies in controlling *C. sinensis* infection.

## Supporting information

**S1 Fig. Prevalence of *Clonorchis sinensis* infection and raw freshwater fish consumption behavior by age and sex in the Republic of Korea, 2012.**
(TIF)

**S2 Fig. Correlation between the prevalence of *Clonorchis sinensis* infection and raw freshwater fish consumption behavior by region in the Republic of Korea, 2012.**
(TIF)

**S1 Table. Details of national intestinal parasitic infections surveys from 1981–2012, the Republic of Korea.**
(DOCX)

**S2 Table. Details of intestinal parasitic infections surveys in residents by five major river streams from 2013–2021, the Republic of Korea.**
(DOCX)

**S3 Table. Model comparisons among all considered age-period-cohort models.**
(DOCX)

**S4 Table. Projected age-stratified prevalence of *Clonorchis sinensis* infection from 2013–2023, Republic of Korea.**
(DOCX)

## Acknowledgments

The authors acknowledge the Korea Disease Control and Prevention Agency and Korea Association of Health Promotion for supplying all available survey reports, which were instrumental in enriching the comprehensive dataset used for all analyses in this study.

## Author Contributions

**Conceptualization:** Sung-mok Jung, Heewon Kang, Jong-hun Kim, Sung Hye Kim.

**Data curation:** Sung-mok Jung, Bong-Kwang Jung, Sejin Ju, Jung-Won Ju, Myoung-Ro Lee, Jong-hun Kim, Sung Hye Kim.

**Formal analysis:** Sung-mok Jung.

**Funding acquisition:** Sung Hye Kim.

**Investigation:** Jong-hun Kim.

**Methodology:** Sung-mok Jung, Heewon Kang.

**Software:** Sung-mok Jung, Heewon Kang.

**Validation:** Sung-mok Jung.

**Visualization:** Sung-mok Jung.

**Writing – original draft:** Sung-mok Jung.

**Writing – review & editing:** Sung-mok Jung, Heewon Kang, Bong-Kwang Jung, Sejin Ju, Jung-Won Ju, Myoung-Ro Lee, Jong-hun Kim, Sung Hye Kim.

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
