## [Decision Letter · Decision Letter 0]

5 Jul 2024

Dear Dr. Jung,

Thank you very much for submitting your manuscript "Age, period, and cohort effects of Clonorchis sinensis infection prevalence in the Republic of Korea: insights and projections" for consideration at PLOS Neglected Tropical Diseases. As with all papers reviewed by the journal, your manuscript was reviewed by members of the editorial board and by several independent reviewers. In light of the reviews (below this email), we would like to invite the resubmission of a significantly-revised version that takes into account the reviewers' comments. 

We cannot make any decision about publication until we have seen the revised manuscript and your response to the reviewers' comments. Your revised manuscript is also likely to be sent to reviewers for further evaluation.

Sincerely,

Ran Wang, M.D.

Academic Editor

Jong-Yil Chai

Section Editor

Reviewer's Responses to Questions

**Key Review Criteria Required for Acceptance?**

**Methods**

-Are the objectives of the study clearly articulated with a clear testable hypothesis stated?

-Is the study design appropriate to address the stated objectives?

-Is the population clearly described and appropriate for the hypothesis being tested?

-Is the sample size sufficient to ensure adequate power to address the hypothesis being tested?

-Were correct statistical analysis used to support conclusions?

-Are there concerns about ethical or regulatory requirements being met?

Reviewer #1: See the attachment.

Reviewer #2: The manuscript did not mention the actual size of investigated population that included in the analysis, which should be provided in the revised version.

Reviewer #3: The 'Age-period-cohort modeling' should be emphasized, including which references were made, how specific calculations were made, and what each coefficient represents.

Reviewer #4: Methods applied to existing data seem appropriate.

As no new data was collected, the ethical considerations of collecting survey data were not necessary to mention.

**Results**

-Does the analysis presented match the analysis plan?

-Are the results clearly and completely presented?

-Are the figures (Tables, Images) of sufficient quality for clarity?

Reviewer #1: See the attachment.

Reviewer #2: Fig 3, there is one blue line, but two shadows (light blue and gray) in each panel.

It seems that gray shadows are not described, suggested to remove them.

Then, you could zoom in on the Y axis.

Reviewer #3: Lines 230-233 The data sources for 2015-2021 should be supported by literature, or relevant tables should be listed in the attachment. Why is there no relevant data for 2013 and 2014?

Reviewer #4: Results are described clearly and appropriately.

**Conclusions**

-Are the conclusions supported by the data presented?

-Are the limitations of analysis clearly described?

-Do the authors discuss how these data can be helpful to advance our understanding of the topic under study?

-Is public health relevance addressed?

Reviewer #1: See the attachment.

Reviewer #2: (No Response)

Reviewer #3: Is there a comparison and discussion between this model and similar models that have already been publicly published?

Reviewer #4: Conclusions support intervention strategies used in the past and in the present. 

Some additional comparison with similar data from China for clonorchiasis would be useful to the reader and further support their conclusions.

**Editorial and Data Presentation Modifications?**

Reviewer #1: Major revisioin.

Reviewer #2: (No Response)

Reviewer #3: Major Revision

Reviewer #4: Well written with no editorial changes suggested.

**Summary and General Comments**

Reviewer #1: 1. Authors has poor knowledge of the distribution of Clonorchis sinensis. Did C. sinensis only endemic to Korea, China and northern Vietnam? In addition, authors made serious mistake. Taiwan is one province of China, so the “Taiwan” should be deleted in this sentence in line 67. Please check and correct.

2. Why use the data from 1981-2012? And what’s the basis? After all, it has been too many years since then.

3. It’s 2024 now. While the study projected the age-stratified prevalence up to 2023 which is not real-time enough, and did not provide the latest data. 

4. What’s the basis for dividing age groups? This should be described. 

5. All tables should be shown as three-line.

Reviewer #2: The manuscript use published data and generate new analysis in particular with the cohort factor. It is a brief research note. Here below are some key points needed to be addressed in the revised version.

1. Line 67, "countries", "China", "Taiwan". When mentioning "countries", it is inproper to put China and Taiwan on the same position, you may use"regions, including... Chinese mainland, Taiwan...". or "countries, including... China (with Taiwan province)...".

as well as at Line298 (e.g., China, Taiwan...)

2. Line 93, "decline in national prevalence to 1.9% by 2012", Is this 1.9% only for C. sinensis?

Maybe the following way is more clear, "...prevalence to 1.9% by 2012 for C. sinensis, comprising 73% of the total positivity rate for all intestinal parasitic infections, the country continues to face areas with high endemicity [5]".

3. In the case of older persons have higher infection proportion, 

did authors exclude the possibility effect of accumulation of raw fish consumption?

4. Discussion should highlight the important effect of widespread allocation of praziquantel since 1983 and the government elimination program since 2005, which reform the trend of going down.

5. Author may discuss: the younger generation (those born in the cohort of 1952 or later) that consuming less raw fish may largely influence by food imported from overseas, and having a keen interest in exploring new eating cultures.

Reviewer #3: This research applied the age-period-cohort modeling approach to nationally representative C. sinensis infection prevalence data from 1981–2012 in Korea, and the model was used to predict the incidence in the next few years. Overall, this study has certain novelty and practical value. However, some questions need to be addressed or answered.

Specific comments

Line 67, Line 298: Please refer to the literature to correctly explain the relations between China and Taiwan（For example, doi: 10.1016/S0140-6736(15)60313-0. doi: 10.1016/j.biotechadv.2018.02.008.）

Fig 3 : What the light and dark shades of blue represent？

Lines 230-233: It was observed that the incidence of Fig3D and E was increased in the high age group, which is not consistent with the result description.

Reviewer #4: The article by Sung-mok Jung et al. explores the use of clonorchiasis survey data to model the impact of age cohorts and location on the prevalence of clonorchiasis in Korea during a successful intervention campaign. The study depends heavily on available datasets that are more than ten years old and they use more recent survey data from high prevalence regions to assess the results of their models. Similar studies have been conducted in China for C. sinensis but similarities in trends observed were not compared/contrasted herein. In particular, how more recent geospatial data in China compares to the correlations observed in prevalance vs. region herein. Some bias in sampling regions with recent comparative data was inevitable due to the reduced prevalance, but should be discussed, perhaps? The results of the article support the importance of considering age group/cohorts when designing intervention strategies. The authors have not provided any changes/modifications from existing intervention methods (education programmes, targeted treatment to reduce parasite load etc.); only that is appears to work well

Other comments:

Author summary is very similar to the abstract. It would be good to review to address accessibillity for a broader audience.

Line 72. Not just transmission to humans

Line 246 247. Is there evidence that age cohort has not been taken into consideration for interventions? It is my understanding that this is known and has been considered in current efforts.

Line 258 to 261. Should also comment on the first intermediate host as well.

Lines 269 to 273. Would modelling low and high risk regions (based on correlation mentioned) be more informative for providing recommendations for future intervention strategies? For example, would the model behave differently in high risk regions?

PLOS authors have the option to publish the peer review history of their article (what does this mean?). If published, this will include your full peer review and any attached files.

Reviewer #1: No

Reviewer #2: Yes: De-Hua Lai

Reviewer #3: No

Reviewer #4: No
---

## [Decision Letter · Decision Letter 1]

23 Sep 2024

Dear Dr. Jung,

Thank you very much for submitting your manuscript "Age, period, and cohort effects of Clonorchis sinensis infection prevalence in the Republic of Korea: insights and projections" for consideration at PLOS Neglected Tropical Diseases. As with all papers reviewed by the journal, your manuscript was reviewed by members of the editorial board and by several independent reviewers. The reviewers appreciated the attention to an important topic. Based on the reviews, we are likely to accept this manuscript for publication, providing that you modify the manuscript according to the review recommendations. 

Sincerely,

Ran Wang, M.D.

Academic Editor

Jong-Yil Chai

Section Editor

Reviewer's Responses to Questions

**Key Review Criteria Required for Acceptance?**

**Methods**

-Are the objectives of the study clearly articulated with a clear testable hypothesis stated?

-Is the study design appropriate to address the stated objectives?

-Is the population clearly described and appropriate for the hypothesis being tested?

-Is the sample size sufficient to ensure adequate power to address the hypothesis being tested?

-Were correct statistical analysis used to support conclusions?

-Are there concerns about ethical or regulatory requirements being met?

Reviewer #1: (No Response)

Reviewer #2: (No Response)

Reviewer #3: (No Response)

**Results**

-Does the analysis presented match the analysis plan?

-Are the results clearly and completely presented?

-Are the figures (Tables, Images) of sufficient quality for clarity?

Reviewer #1: (No Response)

Reviewer #2: (No Response)

Reviewer #3: (No Response)

**Conclusions**

-Are the conclusions supported by the data presented?

-Are the limitations of analysis clearly described?

-Do the authors discuss how these data can be helpful to advance our understanding of the topic under study?

-Is public health relevance addressed?

Reviewer #1: (No Response)

Reviewer #2: (No Response)

Reviewer #3: (No Response)

**Editorial and Data Presentation Modifications?**

Reviewer #1: (No Response)

Reviewer #2: (No Response)

Reviewer #3: Accept

**Summary and General Comments**

Reviewer #1: After carefully reviewing the revised manuscript of the author, it is found that the author has carefully revised the comments and suggestions of the reviewer, only one issue should be address. 

L66-68, “Clonorchis sinensis (C. sinensis)...... including the Republic of Korea (hereafter “Korea”), mainland China, Taiwan, and northern Vietnam [1–3]” should be replaced with “Clonorchis sinensis (C. sinensis), a prevalent intestinal helminth primarily endemic to the East Asian countries, including the Republic of Korea (hereafter “Korea”), China, and Vietnam [1–3]”.

L298, Please delete Taiwan.

It is well known that Taiwan is a province of China, so it is not appropriate to put China and Taiwan together. It is suggested to delete the words "Taiwan" in these sentences.

Please check all text and revise it.

Reviewer #2: (No Response)

Reviewer #3: This study builded and applied the age-period-cohort modeling approach to nationally representative C. sinensis infection prevalence data from 1981–2012 in Korea, in order to provide valuable insights for future public health planning and resource allocation. This research has certain scientific value and reference significance, I agree to publish.

Figure Files:

Data Requirements:

Reproducibility:

References

---

## [Editor Report · Decision Letter 2]

27 Sep 2024

Dear Dr. Jung,

We are pleased to inform you that your manuscript 'Age, period, and cohort effects of Clonorchis sinensis infection prevalence in the Republic of Korea: insights and projections' has been provisionally accepted for publication in PLOS Neglected Tropical Diseases.

Best regards,

Ran Wang, M.D.

Academic Editor

Jong-Yil Chai

Section Editor

---

## [Editor Report · Acceptance letter]

8 Oct 2024

Dear Dr. Jung,

We are delighted to inform you that your manuscript, "Age, period, and cohort effects of Clonorchis sinensis infection prevalence in the Republic of Korea: insights and projections," has been formally accepted for publication in PLOS Neglected Tropical Diseases.

Best regards,

Shaden Kamhawi

co-Editor-in-Chief

Paul Brindley

co-Editor-in-Chief
